# Combined Targeted Omic and Functional Assays Identify Phospholipases A_2_ that Regulate Docking/Priming in Calcium-Triggered Exocytosis

**DOI:** 10.3390/cells8040303

**Published:** 2019-04-02

**Authors:** Deepti Dabral, Jens R Coorssen

**Affiliations:** 1Molecular Physiology and Molecular Medicine Research Group, School of Medicine, Western Sydney University, Campbelltown Campus, NSW 2560, Australia; d.dabral@uws.edu.au; 2Department of Health Sciences, Faculty of Applied Health Sciences and Department of Biological Sciences, Faculty of Mathematics & Science, Brock University, St. Catharines, ON L2S 3A1, Canada

**Keywords:** membrane merger, secretory vesicles, lysolipids, free fatty acids, regulated secretion, fusion

## Abstract

The fundamental molecular mechanism underlying the membrane merger steps of regulated exocytosis is highly conserved across cell types. Although involvement of Phospholipase A_2_ (PLA_2_) in regulated exocytosis has long been suggested, its function or that of its metabolites—a lyso-phospholipid and a free fatty acid—remain somewhat speculative. Here, using a combined bioinformatics and top-down discovery proteomics approach, coupled with lipidomic analyses, PLA_2_ were found to be associated with release-ready cortical secretory vesicles (CV) that possess the minimal molecular machinery for docking, Ca^2+^ sensing and membrane fusion. Tightly coupling the molecular analyses with well-established quantitative fusion assays, we show for the first time that inhibition of a CV surface calcium independent intracellular PLA_2_ and a luminal secretory PLA_2_ significantly reduce docking/priming in the late steps of regulated exocytosis, indicating key regulatory roles in the critical step(s) preceding membrane merger.

## 1. Introduction

Fusion is a ubiquitous fundamental cellular process enabling merger of biological membranes; this includes the defining release step of exocytosis following merger of secretory vesicles with the plasma membrane (PM). Studies over the last three-four decades have suggested a role of Phospholipase A_2_ (PLA_2_) in regulated exocytosis, generally arising from correlations of arachidonic acid (ARA) levels and vesicle content release [1,2,3,4,5,6,7,8]. In a cell free system, fusion of secretory vesicles with target membranes increased in the presence of PLA_2_ suggesting that the enzyme or its metabolites affected membrane merger or the steps immediately preceding it [5,6,9]. Furthermore, addition of exogenous PLA_2_ metabolites inhibits hemi-fusion formation or transition from hemi-fusion to pore opening depending upon their site of incorporation [10,11,12]. That PLA_2_ species have been identified in secretory vesicles isolated from neutrophils [13], chromaffin cells [14], insulin secreting cells [15,16] and mast cells [17,18] highlights that these are conserved protein components of secretory vesicles. Early studies also identified endogenous PLA_2_ activity in purified synaptic vesicles (SV) [5,6,19,20]. While some proteomic analyses have failed to identify SV associated PLA_2_ [21,22], suggesting very low abundance and/or activity, at least one study identified iPLA2γ in synaptosomes [23]. Indeed, when synaptosomes were fractionated into free and docked SV pools prior to proteomic analysis, a patatin-like phospholipase domain containing 8 (Gene ID 157819923; also known as iPLA_2_γ) was found to be more abundant in the docked SV pool [24].

Based on subcellular localization and Ca^2+^ requirement, PLA_2_ isozymes are broadly classified as (i) Ca^2+^-dependent secretory or sPLA_2_, (ii) Ca^2+^-independent cytoplasmic or membrane bound iPLA_2_ and (iii) Ca^2+^-dependent cytoplasmic or cPLA_2_. Nevertheless, all PLA_2_ cleave membrane phospholipids at the sn-2 position, producing a lyso-phospholipid (often lyso-phosphatidylcholine; LPC) which is preferentially retained in the membrane and a diffusible free fatty acid (FFA; often arachidonic acid (ARA; 20:4n-6)) that can incorporate back into the membrane via Lands cycle intermediates or is further metabolized to yield eicosanoids. sPLA_2_ are generally 12–19 kDa proteins containing 5–7 conserved disulphide bonds and are characterized by an active site histidine in a DXCCXXHD consensus sequence and a conserved Ca^2+^-binding loop with a XCGXGG consensus sequence [25,26]. Once secreted, their Ca^2+^ requirement for catalysis is in the low millimolar (mM) range with no strict fatty acid selectivity, although mostly target phosphatidyl-ethanolamine, -glycerol, -inositol and -serine (PE, PG, PI and PS) [26,27]. Although these enzymes are generally thought to be active only when released into the extracellular space, the possibility of basal sPLA_2_ activity in the vesicle lumen has been suggested [17,28,29]. In transfected CHO-K1 and HEK293 cells that lack regulated secretory vesicles, expressed sPLA_2_ was localized to the golgi and related intracellular vesicles and released ARA, indicating basal activity [28]. Of note is that adequate concentrations of free calcium ([Ca^2+^]_free_) may be (transiently) present within regulated secretory vesicles to support luminal sPLA_2_ activity under basal conditions [30,31,32,33]. In contrast, iPLA_2_ are intracellular enzymes that do not require Ca^2+^ beyond basal cytosolic levels for activity, contain serine in the GXSXG active site and possess a nucleotide-binding (GXGXXG) consensus sequence. The mammalian forms include iPLA_2_β, iPLA_2_γ, iPLA_2_δ, iPLA_2_ε, iPLA_2_ζ and iPLA_2_η with molecular weights ranging from 27 to 146 kDa. A well-established function of iPLA_2_β is membrane phospholipid remodelling via Lands cycle to maintain membrane integrity [29,34,35]. cPLA_2_ are cytosolic proteins of molecular weights ranging from 61-114 kDa and require micromolar [Ca^2+^]_free_ for association with membrane phospholipids via their N-terminal C2 domain [25,26]. These enzymes also carry serine at GXSXG, GXSGS or GXSXV lipase consensus sequences [36]. There are six mammalian cPLA_2_ isoforms, cPLA_2_-α, β, γ, δ, ε, ζ of which all except cPLA_2_γ have a N-terminal C2 domain [36,37].

At the genomic level, sea urchin and human remain closely related (both are deuterostomes) relative to invertebrates [38,39,40]. SNARE (soluble N-ethylmaleimide-sensitive factor attachment protein receptor) proteins are highly conserved across species, having been identified on exocytotic vesicles from yeast to man, including the release-ready cortical vesicles (CV) isolated from unfertilized sea urchin eggs [40,41,42,43,44,45,46]. Furthermore, as CV retain the minimal molecular machinery for docking, Ca^2+^ sensing and membrane fusion, they are a well-established model to study the late steps of Ca^2+^-triggered exocytosis [42,43,45,47,48,49,50,51,52,53,54]. CV are easily isolated with high purity and yield, either endogenously docked on the PM (i.e., cell surface complexes; CSC) or free-floating [42,49,51,55]. On triggering with Ca^2+^, isolated CV fuse with each other in a manner indistinguishable from CV-PM fusion [42,45,49,55]. Using these fusion-ready native vesicles, we first identified catalytically active endogenous PLA_2_ isozymes using a combined bioinformatics and top-down discovery proteomics approach coupled with lipidomic analyses. We then asked whether these isozymes have a role in the late steps of Ca^2+^-triggered exocytosis. The enzymatic activities were localized to the CV lumen and vesicle surface and tightly coupled functional and molecular analyses were carried out to assess these PLA_2_ activities and their associated effects on fusion parameters. A modulatory role in priming and/or docking is indicated.

## 2. Materials and Methods

### 2.1. Materials

#### 2.1.1. Assessment of PLA_2_ Activities and Fusion Assays

Native sea urchins (Heliocedaris tuberculata) were maintained at 7–8 °C in the local aquatic facility. CSC and CV were isolated as previously described [38,41,42,45,51,52,55]. Bromoenol lactone (BEL) and 3-(3-acetamide-1-benzyl-2-ethylindolyl-5-oxy) propane sulfonic acid (LY311727) were purchased from Sigma Aldrich (St. Louis, MO USA); 1,1,1,2,2-Pentafluoro-7-phenyl-3-heptanone (FKGK-11) was from Cayman Chemicals (Ann Arbor, MI, USA).

CSC membrane labelling was carried out using 1-oleoyl-2-{12-[(7-nitro-2-1,3-benzoxadiazol-4-yl) amino] dodecanoyl} -sn-glycero-3-phospho-choline/-ethanolamine (NBD-PC/PE). Quantification of the generated NBD-FFA and NBD-PA was done using 1-oleoyl-2-{12-[(7-nitro-2-1,3-benzoxadiazol-4-yl) amino Arachidonic acid 20:4 (NBD-ARA) and 1-oleoyl-2-{12-[(7-nitro-2-1,3-benzoxadiazol-4-yl) amino] dodecanoyl}-sn-glycero-3-phosphate (NBD-PA) as reference standards. ARA (20:4) was purchased from Matreya LLC (State college, PA, USA) while all other lipids were purchased from Avanti Polar Lipids (Alabaster, AL, USA). Quantitative lipid analyses were carried out using High Performance Thin Layer Chromatography (HPTLC) [49,50,51,56]. Silica G-60 plates were purchased from Merck Millipore Ltd. (Darmstadt, Germany). All HPTLC solvents were minimally of analytical grade. High grade Trypsin was from Promega Corp. (Madison, WI, USA). The selective PLA_2_ substrate, PED6 (*N*-((6-(2,4-Dinitrophenyl) amino) hexanoyl)-2-(4,4-Difluoro-5,7-Dimethyl-4-Bora-3a,4a-Diaza-s-Indacene-3-Pentanoyl)-1-Hexadecanoyl-sn-Glycero-3-Phosphoethanolamine, Triethylammonium Salt) was from Invitrogen (Carlsbad, CA, USA).

#### 2.1.2. Protein Fractionation, Gel Electrophoresis and Western Blotting

Amicon Ultra-4 centrifugal filters (3kDa cut-off) came from Merck Millipore. Constituents of the Protease inhibitor cocktail [41,57] were purchased from Sigma-Aldrich (St. Louis) and AMRESCO Inc. (Dallas, TX, USA). All electrophoresis grade chemicals used in two-dimensional gel electrophoresis (2DE) were purchased from AMRESCO Inc. 3-10NL IPG strips, tributyl phosphine and Polyvinylidene difluoride (PVDF) membrane (0.2 µM pore size) were from Bio-Rad (Hercules, CA, USA). Anti-human rabbit polyclonal iPLA_2_ antibody (PA5-27945) and anti-human mouse monoclonal sPLA_2_ antibody (ab-24498) were from Invitrogen and Abcam (Cambridge, UK), respectively. Blocking agents―non-fat dry milk powder and BSA―were from Devondale (Saputo Dairy, Australia) and Sigma Aldrich, respectively. HRP linked anti-mouse IgG (NA93IV) and anti-rabbit IgG (NA934VS) antibodies came from GE Healthcare (Buckinghamshire, UK). Lumunata Cresendo Western HRP substrate was purchased from Merck Millipore. The blots and gels were imaged using the Image Quant™ LAS 4000 Biomolecular Imager (GE Healthcare, Chicago, Il, USA).

### 2.2. Methods

#### 2.2.1. Molecular Analysis to Detect CSC Associated PLA_2_ Activities

PLA_2_ activities in intact unlabelled CSC suspensions (0.5 mL, OD 0.56 ± 0.02, *n* = 3–14) were assessed by measuring changes in FFA following treatments with the indicated concentrations of BEL, LY311727 and FKGK-11; exploratory experiments (*n* = 1–2) with 40 µM BEL, 10 µM FKGK-11, 100 nM LY311727 and higher concentrations were first carried out. Only those concentrations that significantly reduced endogenous FFA relative to control were selected for further study. CSC were isolated according to established protocols [49,50,51,52] and suspended in baseline intracellular medium (BIM; 210 mM potassium glutamate, 500 mM glycine, 10 mM NaCl, 10 mM PIPES, 50 μM CaCl_2_, 1 mM MgCl_2_, 1 mM EGTA pH 6.7) supplemented with 2.5 mM ATP, 2 mM DTT and 1X protease inhibitors [41,42,43,45,51,52]. Stock solutions of inhibitors were prepared in dimethyl sulfoxide and were delivered to CSC suspensions at a final solvent concentration of <1% [48,51,52]. At the end of the incubation, CV suspensions were immediately placed on ice, diluted with 5–8 mL of ice-cold BIM and aliquoted for rapid isolation of total lipids and for immediate assay of CV-PM fusion [49,50,51].

In separate experiments, CSC were allowed to incorporate NBD-PC or -PE for 15 min prior to the 20 min inhibitor treatments. Briefly, CSC suspensions (7 mL, OD 0.56 ± 0.005, *n* = 3) were labelled with 5600 picomoles NBD-PC or -PE for 15 min followed by treatment with the indicated doses of inhibitors. At the end of the treatments, all aliquots were washed three times in ice-cold BIM (pH 6.7) to remove unlabelled NBD substrate and inhibitors and immediately placed on ice prior to rapid isolation of total CSC lipids and membrane proteins (see Section 2.2.4).

#### 2.2.2. Quantitative Lipid Analyses

Total CSC or CV lipids were extracted according to Bligh and Dyer with established modifications [50,51,52,56,58]; total CV membrane proteins were also isolated from parallel aliquots by lysing CV with ice-cold PIPES (see Section 2.2.4). The resulting organic phases (and also aqueous phases in scaled-up experiments) were recovered, dried under nitrogen and suspended in CHCl_3_:CH_3_OH (2:1; *v*/*v*) for loading onto pre-conditioned silica gel 60 HPTLC plates. Sample loading was with the CAMAG automatic TLC sampler. Pre-conditioning of the plates was as previously described [50,51,52,56]. Using the CAMAG AMD 2 multi-development unit, neutral- and phospho-lipids were resolved using optimized protocols [48,49,51,52,56]. Resolved lipids were visualized on-plate by charring homogenously wetted plates with 10% cupric sulphate in 8% aqueous phosphoric acid at 145 °C for 10 min [56]; imaging was at 460 nm/605 nm (Ex/Em) using the LAS 4000 Biomolecular Imager and was analysed using MultiGauge v3.0 (Fuji Photo Film Co., Ltd. Tokyo, Japan). Quantification of endogenous CV or CSC lipids, NBD-FFA and -PA, was via calibration curves of the appropriate standards, resolved by HPTLC, in parallel with the experimental samples. To resolve neutral and phospholipids, well-established protocols were used [56]; NBD-FFA and -PA generated from NBD-PC labelled CSC were resolved together after confirming that the standard neutral lipid protocol was satisfactory for resolving both these species. To measure fluorescent signal from NBD-labelled lipids, uncharred plates were imaged at 460 nm/575 nm (Ex/Em) using the same imager and analysis software.

#### 2.2.3. Sequence Alignments

To identify regions of homology, a sequence alignment of the putative urchin sPLA_2_ (W4XE93) and iPLA_2_ (XP_011669048.1) relative to characterized human and mouse isoforms was carried out using MEGA version 7.0.21 (free software downloaded from https://www.megasoftware.net/home) [59,60] and the ClustalW algorithm with default settings. All protein sequences, except that of the predicted urchin iPLA_2_, were downloaded from Uniprot (February 2019). Of 22 putative urchin sequences, one with sPLA_2_ characteristics (i.e., ~14kDa, likely to form intra-molecular disulphide bonds and prediction of Ca^2+^ as a cofactor [17,25]) was selected using the information available in Uniprot. Similarly, a putative urchin 80–85kDa iPLA_2_ was selected from the 44 entries on the basis of characteristic molecular weight and presence of ankyrin repeats—a feature common to iPLA_2_ forms across species [25,35]. As the predicted urchin 80–85kDa iPLA_2_ is not present in the Uniprot database, the sequence was downloaded from PubMed. Human PLA_2_ sequences used for the analyses were sPLA_2_–P14555 and iPLA_2_–O60733 and murine PLA_2_ sequences were sPLA_2_–Q9WVF6 and iPLA_2_–P97819. Only those human and murine PLA_2_ sequences identified at the protein level were selected. As the predicted urchin sequences present in Uniprot or PubMed (66 sequences in total) do not contain the C2 domain characteristic of cPLA_2_, sequence alignment of cPLA_2_ isozymes was not carried out and there was no further testing for this isozyme. Building from the in-silico analysis, validation at the protein level was then carried out using a well-established top-down proteomic approach coupling 2DE with high-sensitivity western blotting using anti-human sPLA_2_ and iPLA_2_ antibodies to conserved regions [45,57,61].

#### 2.2.4. Sample Fractionation for 2DE Western Blotting

After de-jellying and thorough washing, unfertilized urchin eggs were homogenized in intracellular media (IM; 220 mM K-glutamate, 500 mM glycine, 10 mM NaCl, 5 mM MgCl_2_, 5 mM EGTA, pH 6.7) supplemented with 2.5 mM ATP, 1 mM benzamidine HCl, 2 mg/mL aprotinin, 2 mg/mL pepstatin, 2 mg/mL leupeptin and 2 mM DTT) prior to standard isolation of high purity CSC or CV suspended in BIM [38,42,45,47,48,49,51,52]. For fractionation, CV were lysed using three volumes of ice-cold PIPES buffer (20 mM PIPES, pH 7.0 supplemented with protease, phosphatase and kinase inhibitors) for 90 s and isotonicity restored by adding an equal volume of 2 X IM (supplemented with protease, phosphatase and kinase inhibitors). Samples were centrifuged at 125,000× *g*, 4 °C, for 3 h to recover membrane pellets and CV luminal proteins in the supernatant [50]. Membranes were solubilized in 2DE buffer (4% CHAPS, 8 M Urea and 2 M Thiourea supplemented with protease, phosphatase and kinase inhibitors [50,57]) while supernatant containing CV luminal proteins was buffer exchanged with 3M urea solution using Amicon filters. Solubilized membrane proteins and concentrated CV luminal proteins were resolved using a well-established 2DE protocol involving isoelectric focusing (IEF) followed by SDS-PAGE in 1 mm thick 10% resolving gels [57]. Subsequent high sensitivity western blotting was essentially as previously described [41,42,45,61]. Briefly, resolved proteins were transferred to 0.2 µM pore size PVDF membrane (Bio-Rad) using buffer containing 25 mm Tris, 192 mm glycine, 20% methanol and 0.025% SDS at 90 Volts for 1 h. Blots were blocked for 1 h at room temperature (RT) in a solution containing 20 mM Tris, 150 mM NaCl, 0.1% Tween (TBST) and 5% non-fat dry milk or 20 mg/mL BSA. Blots were incubated overnight (at least 16 h) with constant rocking at 4 °C with anti-human sPLA_2_ (1:800) or iPLA_2_ (1:2000) primary antibodies. After washing, blots were incubated with secondary antibodies conjugated to horseradish peroxidase (HRP) for 2 h at RT and then detected using Luminata Crescendo western HRP substrate and the LAS 4000 Biomolecular Imager. Secondary antibody controls were always carried out in parallel western blots and parallel 2DE gels were stained with colloidal Coomassie Brilliant Blue (cCBB) [62,63].

#### 2.2.5. Molecular Analyses to Detect CV Associated PLA_2_ Activities

Membrane and luminal fractions from purified CV suspensions were isolated using the same overall strategy as above except for the use of ice-cold BIM (pH 6.7) in the last step to maintain endogenous PLA_2_ activity. This also ensured tight coupling of functional (i.e., fusion) and molecular analyses. PLA_2_ activity in CV membrane and lumen fractions was measured by incubating 30 µg CV membrane or luminal protein with 200 picomoles NBD-PE or 170 picomoles NBD-PC for 30 s–10 min. PLA_2_ activity was then stopped by adding 0.2 µL HCl and all aliquots were immediately placed on ice for rapid isolation of total lipids; scaled-up experiments were carried out using more CV luminal protein (120 µg) and NBD-PE (664 picomoles) to best detect NBD-FFA species. An equal volume of BIM without CV luminal or membrane protein was used as a control to which the indicated amount of NBD-PC or -PE was added and incubated for 10 min.

To further establish whether catalytically active PLA_2_ was on the outer CV monolayer, well-established trypsin treatments were carried out to remove CV surface proteins and PLA_2_ activity was measured using the selective PLA_2_ substrate, PED6. Briefly, CV suspensions (2–4 mL; OD 0.95 + 0.04; *n* = 3) were treated with 700 U/mL trypsin for 1 h with gentle mixing every 15 min [42,45]. After treatment, CV suspensions were aliquoted into microplates and incubated with PED6 for 0–30 min; PLA_2_ activity was measured at 485 nm/520 nm (Ex/Em) using a POLARstar Omega microplate reader. Fusion assays were also carried out to confirm the fusion competence of the treated CV. To confirm removal of iPLA_2_ from the CV surface, western blotting was carried out on CV membrane proteins using the iPLA_2_ antibody. Protein isolation and SDS-PAGE were as above with the exception that SDS sample buffer was used for solubilizing CV membrane proteins and detection was with cCBB [41,42,45,62,63]. In parallel replicates, absorbance at A405 was measured regularly during the incubations to assess for possible CV lysis; any drastic decrease in absorbance indicated marked loss of CV due to bursting (as seen with the addition of ddH_2_O at the end of the experiment to confirm the intact state of CV).

#### 2.2.6. Effects of PLA_2_ Activities on Fusion Assays

Standard fusion and docking/priming assays were carried out as previously described [42,43,45,47,48,49,50,51,52,54,56]. All experiments were carried out in BIM pH 6.7 supplemented with 2.5 mM ATP, 2 mM DTT and 1X protease inhibitors [41,42,43,45,51,52]. Briefly, CSC (0.5 mL, OD 0.56 ± 0.02, *n* = 3–14) and CV (OD 0.94 ± 0.03, *n* = 22) suspensions were treated with the indicated inhibitor concentrations. Immediately following treatments, suspensions were diluted with 3–5 volumes of ice-cold BIM, pH 6.7 and aliquoted into microplates. Each condition was challenged with increasing concentrations of [Ca^2+^]_free_ in quadruplicate, consistent with established protocol and this was repeated in separate experiments as indicated (*n*) [42,43,45,47,48,49,50,51,52]. The decrease in OD representing CV fusion was measured using a POLARstar Omega microplate reader [53,54]. Final [Ca^2+^]_free_ were measured with a Ca^2+^ sensitive electrode (Calcium Combination ISE, EDT directION Limited) as previously described [53,54]. All treatments were carried out at 25 ℃. Parallel solvent controls were carried out in every experiment. A separate aliquot in each experiment was also snap-frozen in liquid N_2_ and stored at −80 ℃ for subsequent molecular analyses.

## 3. Data Analyses

All data are reported as mean ± S.E.M. Two-sample two-tailed t-tests and 2-way ANOVA with Tukey’s multiple comparisons were performed using GraphPad PRISM 7 version 7.04 to assess the difference between experimental conditions (*p* ≤ 0.05 are considered significant).

## 4. Results

### 4.1. Identification of Endogenous PLA_2_ Activities on CSC and Their Effects on CSC Fusion

As CSC consist of CV endogenously docked on the PM and undergo Ca^2+^-triggered exocytosis in vitro, the potential presence and influence(s) of PLA_2_ activities was first assessed using CSC and selective PLA_2_ inhibitors. Treating CSC with LY311727 and BEL caused significant concentration-dependent decreases in endogenous FFA levels relative to a basal FFA level of 267.8 ± 29.8 femtomoles per µg CSC membrane protein (fmoles/μg MP) (Figure 1A; *n* = 3); 200 µM LY311727 reduced FFA levels to 122.7 ± 20.13 fmoles/μg MP while 100 µM and 500 µM BEL treatments reduced FFA levels below background values. No significant change in FFA levels was observed in any of the FKGK-11 treatments. As the lipid detection approach used is somewhat selective for unsaturated lipid species [56], in order to ensure a full assessment of potential PLA_2_ activities, CSC were labelled with NBD-PC (fluorophore at the sn-2 fatty acid); as previously documented, such exogenous substrates are readily incorporated into CSC and CV membranes [49]. The generation of 3.8 ± 1.2 picomoles NBD-FFA/μg MP from hydrolysis of the incorporated NBD-PC further confirmed endogenous PLA_2_ activity (Figure 1B; *n* = 3). The NBD-FFA was also generated from NBD-PE labelled CSC and CV (*n* = 1 confirmatory experiment, not shown). However, in contrast to Figure 1A, treating NBD-PC labelled CSC with increasing doses of BEL resulted in a concentration-dependent increase in NBD-FFA to 7.2 ± 3.0 and 9.8 ± 2.9 picomoles/μg MP. Similarly, a concentration-dependent increase in NBD-FFA to 8.9 ± 6.4 and 11.5 ± 8.2 picomoles/μg MP was observed in NBD-PC labelled CSC treated with 50 µM and 200 µM FKGK-11, respectively. No significant change in NBD-FFA levels were observed in any of the LY311727 treatments. As PLD also acts on PC, generation of 2.8 ± 0.5 picomoles NBD-PA/μg MP confirmed endogenous PLD activity (Figure 1B) consistent with previous findings [49]. However, none of the inhibitors caused significant change in NBD-PA with respect to the control. Importantly, the decrease in endogenous FFA levels (Figure 1A) and an increase in endogenous TAG, the *de novo* metabolite ([50,56]; Appendix A and also see [34,64,65]) in the BEL treated unlabelled CSC indicated the presence of a BEL sensitive PLA_2_ on CSC. However, the significant increase in NBD-FFA levels in BEL and FKGK-11 treated NBD-PC labelled CSC also indicated the presence of a BEL insensitive PLA_2_ species (Figure 1B). As BEL and FKGK-11 inhibit iPLA_2_ and LY311727 is a selective inhibitor of sPLA_2_ [26,35,64,66,67,68], the data suggested the presence of different catalytically active endogenous PLA_2_ in CSC preparations.

With the evidence of active endogenous PLA_2_, the effects of selective PLA_2_ inhibitors were then tested on CV-PM fusion. The native, docked CV-PM preparations exposed to increasing [Ca^2+^]_free_ undergo exocytosis in vitro yielding a classic sigmoidal Ca^2+^ activity curve with an EC50 of 38.9 ± 6.4 µM [Ca^2+^]_free_ (Figure 2; *n* = 14) [42,45,48,49,52,69,70]. Treating CSC with 20 µM and 200 µM LY311727 [66,68] significantly decreased fusion extent to 44.1 ± 5.3% and 33.6 ± 20.8%, respectively (Figure 2; *n* = 3–4). On treating CSC with 100 µM and 500 µM BEL [35,64,71,72] a significant decrease in the fusion extent to 53.7 ± 24.6% and 54.2 ± 7.9%, respectively, was observed; 500 µM BEL also caused a progressive rightward shift in Ca^2+^ sensitivity to an EC50 of 230.6 ± 101.5 µM [Ca^2+^]_free_ (Figure 2; *n* = 3). CSC treated with 50 µM and 200 µM FKGK-11, another iPLA_2_ inhibitor [67], also significantly reduced fusion extent to 71.4 ± 28.5% and 31.3 ± 23.6%, respectively (Figure 2; *n* = 3). Although, LY311727, BEL and FKGK-11 inhibited endogenous FFA production and blocked triggered fusion; PLA_2_ species inhibited by LY311727 seem separate from the species which was inhibited by BEL and FKGK-11, in particular with reference to NBD-FFA generated from exogenous NBD-PC.

### 4.2. In-Silico Analysis

With initial evidence indicating the likely presence of more than one PLA_2_ isozyme on CSC and their potential role in regulating late steps of regulated exocytosis, we sought to identify these isozymes, beginning with sequence alignment to identify regions of homology in putative urchin PLA_2_ and characterized human and murine forms. The sequence alignment of sPLA_2_ and iPLA_2_ in the selected species clearly indicated conserved amino acid residues throughout the primary amino acid sequences, consistent with a high overall conservation from urchin to human (Figure 3) [40,73,74]. The urchin sPLA_2_ and iPLA_2_ forms showed high identity and similarity with both the human and murine species, particularly around the catalytic site histidine with >75% identity and similarity, as well as around the catalytic site serine with 100% identity and similarity (Figure 3A,B and Table 1). The sPLA_2_ also have a histidine in the Ca^2+^ binding loop with a XCGXGG consensus sequence to which LY311727 binds to inhibit the enzyme [66,68]. The putative urchin iPLA_2_ also contains a nucleotide binding site with a characteristic GGGVKG consensus sequence [35,75] and a caspase catalytic site with the DVTD consensus sequence [35,76,77]. Cysteine residues at C-67, C-185, C-240, C-344, C-465, C-550 and C-821 in the putative urchin iPLA_2_ are also conserved and are thus likely susceptible to alkylation during BEL treatments [78].

### 4.3. Western Blotting and Assays to Confirm Identity and Localization of PLA_2_ Isozymes

After identifying conserved catalytic and regulatory domains in the putative urchin isozymes (Figure 3 and Table 1), enzyme localization in CSC and CV was probed using selective antibodies. The anti-human sPLA_2_ antibody was raised against the entire length of human sPLA_2_ and would thus identify putative urchin sPLA_2_ considering the high identify and similarity, particularly within the catalytic region (Figure 3A and Table 1). Similarly, the anti-human iPLA_2_ recognizes a region between amino acid 399–704, which is also highly conserved in the putative urchin iPLA_2_ isozyme (54% identify, 67% similarity); this region also carries highly conserved GXSXG and GGGVKG catalytic and nucleotide binding consensus sequences (Figure 3B and Table 1). Western blotting of 2D gels identified two immune positive spots of ~14 kDa and pI of ~4.0 and ~4.9 in the CV lumen fraction using the sPLA_2_ antibody (Figure 4A). An immune positive spot of ~63 kDa and pI ~6.2 and two immune positive spots of ~63 kDa and pI ~5.2 and 5.4 were detected in the CSC and CV membrane fractions, respectively, using the iPLA_2_ antibody (Figure 4B,C, respectively). Of note is the relative high abundance of iPLA_2_ on the CSC membrane in comparison to the CV membrane.

To further and more specifically localize the PLA_2_ isozymes and explore possible substrate preferences, NBD-PE and -PC were used to assess the CV associated enzyme activities. First, an isolate of CV lumen proteins was incubated with NBD-PE or -PC for different interval (Figure 5A; *n* = 3). The NBD-PE in controls after 10 min incubation was 12.3 ± 0.9 picomoles which was significantly reduced to 9.0 ± 1.2, 7.4 ± 1.2, 6.8 ± 1.0 and 3.8 ± 2.3 picomoles following 30 s, 2 min, 5 min and 10 min incubations, respectively (Figure 5A). The generation of NBD-FFA having a retention factor (Rf) similar to NBD-ARA indicated PLA_2_ activity; an ~70–90% increase in NBD-FFA over time was observed (Figure 5A; *n* = 2–3). Notably, a very large increase in NBD-FFA in CV aliquots that were pre-incubated with 104.0 µM [Ca^2+^]_free_ confirmed Ca^2+^ promoted CV luminal PLA_2_ activity (Figure 5B, *n* = 3). In contrast, 18.4 ± 2.0 picomoles NBD-PC was detected in buffer controls and no significant change was detected even after 10 min incubation with the CV luminal protein (Figure 5A). This correlated with no detection of NBD-FFA species in NBD-PC labelled CSC (*n* = 2 confirmatory experiments, not shown). Thus, a decrease in NBD-PE and parallel increase in NBD-FFA over time and a large increase in the generation of NBD-FFA in the presence of higher [Ca^2+^]_free_ confirmed the presence of a Ca^2+^ dependent catalytically active sPLA_2_ in the isolated CV luminal fraction, which is selective for PE over PC.

The substrate preference of the ~63kDa CV membrane iPLA_2_ was also investigated by incubating isolated CV membrane fragments with NBD-PE or -PC (Figure 6A; *n* = 2–3). Buffer without CV membrane was used as the control to which the indicated amount of NBD-PC or -PE was added and incubated for 10 min. In all incubations, no significant change in the NBD-labelled lipids nor generation of NBD-FFA was observed. Therefore, to confirm whether the apparent membrane associated iPLA_2_ was lost or became inactive during CV membrane isolation, intact CV were assayed for PLA_2_ activity following trypsin treatment and loss of iPLA_2_ was confirmed by western blotting. Loss of the ~63 kDa iPLA_2_ protein band following trypsin treatment confirmed the presence of iPLA_2_ on the CV surface (Figure 6B, *n* = 3; see also Appendix A). Both control and trypsin treated CV were then incubated with the PLA_2_ selective substrate PED6—a glycerophosphoethanolamine with a BODIPY dye-labelled sn-2 acyl chain and a dinitrophenyl quencher-modified head group [79,80]. The cleavage of the BODIPY dye from the sn-2 position by the action of PLA_2_ resulted in decreased quenching by the head group which specifically indicated PLA_2_ activity. Relative to the 0 min control, there was no detectable PLA_2_ activity in the trypsin treated CV, until 2 min. No significant change in the fluorescence intensity between 2–30 min in trypsin treated CV, relative to 0 min, indicated a complete loss of iPLA_2_ activity that correlated with loss of the iPLA_2_ band on the western blots (Figure 6C; *n* = 3). Absorbance measurements of parallel samples confirmed that there was no major loss of CV over the course of these incubations (not shown).

Considering the apparent complete loss of the ~63 kDa iPLA_2_ from the external membrane of intact CV following trypsin treatment and the corresponding loss of activity assessed using the selective PED6 substrate (Figure 6B,C), we also tested these same batches of control and trypsin treated CV for functional activity. Parallel standard and settle fusion assays were thus carried out (Figure 6D,E, *n* = 2–3). In the standard assay, there was no significant change in the fusion extent although Ca^2+^ sensitivity appeared right-shifted following trypsin treatment; however, the shift was not significant relative to the untreated controls. Notably, in the settle assay used to assess docking and priming [42,47,49,50,51,52,54], fusion extent was significantly decreased to 54.3 ± 7.2% following trypsin treatment (Figure 6E; *n* = 3, (*p* < 0.004)).

### 4.4. The Effect of PLA_2_ Activities on CV-CV Fusion

As in previous work, to enable a tight focus on the late steps of Ca^2+^ -triggered exocytosis, including the fusion mechanism itself, without having to contend with the ‘background’ of the PM, high purity CV preparations were used to further assess the potential influences of the endogenous CV-associated PLA_2_ activities. Both the sPLA_2_ and iPLA_2_ were thus again targeted with selective inhibitors to better assess their potential influence(s) on the late steps of regulated exocytosis (Figure 7A,B; *n* = 3–19). The standard endpoint CV-CV fusion assays yielded a characteristic sigmoidal Ca^2+^ activity curve with an EC50 of 44.8 ± 3.8 µM [Ca^2+^]_free_ (Figure 7A; *n* = 19), which was translationally invariant with CV-PM fusion (Figure 1; *n* = 14) [42,45,49,51]. A significant decrease in fusion extent to 95.3 ± 2.9% and 80.0 ± 10.7%, was observed in CV treated with 20 µM and 200 µM LY311727, respectively (Figure 7A). Fusion of CV treated with 100 µM and 500 µM BEL were also reduced to 96.0 ± 3.2% and 90.0 ± 6.4%, respectively (Figure 7A); following treatment with 50 µM and 200 µM FKGK-11 fusion extent was reduced to 87.4 ± 8.5% and 75.4 ± 7.3%, respectively (Figure 7A). A significant right-shift in Ca^2+^ sensitivity, to yield EC50 of 78.9 ± 9.4 µM [Ca^2+^]_free_ and 79.8 ± 7.5 µM [Ca^2+^]_free_ was also observed following treatment with 50 µM and 200 µM FKGK-11.

When tested in the settle assay, CV treated with all doses of the inhibitors showed further decreases in fusion extent. These concentration-dependent decreases in fusion were to 89.9 ± 4.6% and 53.5 ± 13.4% in 20 µM and 200 µM LY311727 treated CV, respectively; 91.3 ± 4.6% and 87.6 ± 6.6% in 100 µM and 500 µM BEL treated CV, respectively; and 77.0 ± 16.0% and 58.9 ± 10.9% in 50 µM and 200 µM FKGK-11 treated CV, respectively, confirming the role(s) of PLA_2_ isozymes in promoting and maintaining CV priming and/or docking (Figure 7B). To confirm that changes in the CV-CV fusion parameters were associated with perturbed PLA_2_ activities, total lipid changes were assessed from the same preparations following inhibitor treatments. Treating CV with LY311727, BEL and FKGK-11 caused significant concentration-dependent decreases in endogenous FFA levels, relative to a basal FFA level of 15.7 ± 0.51 fmoles/μg MP (Figure 8; *n* = 3–4). FFA levels decreased to 12.9 ± 1.3 and 8.9 ± 3.5 fmoles/μg MP in CV treated with 20 µM and 200 µM LY311727, respectively. Additionally, a significant increase in the endogenous PE levels in the CV treated with 200 µM LY311727 (Table 2), confirmed the inhibitory action of LY311727 on PE-selective PLA_2_. Treating CV with 500 µM BEL also significantly reduced FFA levels to 1.3 ± 6.7 fmoles/μg MP (Figure 8A,B; *n* = 4–12) and increased TAG levels to 376.7 ± 22.5 picomoles/μg MP, relative to the basal level of 217.2 ± 40.9 picomoles in the control (Appendix A, *n* = 3–8; also seen in the BEL treated CSC, Appendix A, *n* = 3–5).

## 5. Discussion

Using bioinformatics and a top-down proteomic approach combined with lipid analyses of isolated native CSC and CV, here we have identified endogenous, vesicle associated PLA_2_—a sPLA_2_ in the CV lumen and an iPLA_2_ on the CV surface—and also demonstrated that their activities play a role in promoting and maintaining the docked and/or primed state of secretory vesicles. The results are consistent with previous findings in platelets that PLA_2_ activity is not essential for fusion *per se* but has clear modulatory roles late in the exocytotic pathway [81].

### 5.1. PLA_2_ Activity and Their Effects on CSC Fusion

Initial assessments of CSC confirmed the existence of endogenous PLA_2_ activities and that these were associated with the late steps of the regulated exocytotic pathway (Figure 1 and Figure 2). Blockade of these PLA_2_ activities using the well-characterized iPLA_2_ inhibitors, BEL [35,64,66,78] and FKGK-11 [35,67] and the sPLA_2_ inhibitor LY311727 [68], caused significant decreases in endogenous FFA levels (Figure 1A) and an increase in PE levels (Table 2). Importantly, a significant decrease in endogenous FFA but increase in NBD-FFA in BEL treated CSC also indicated the presence of BEL sensitive as well as insensitive PLA_2_ isozymes, the latter seemingly an sPLA_2_ (Figure 1A,B). Furthermore, to assess for potential nonspecific inhibitory effects of BEL on other enzymes [78], unlabelled and NBD-PC labelled CSC were treated with another iPLA_2_ inhibitor, FKGK-11. The significant concentration-dependent increase in NBD-FFA (similar to that seen with BEL; Figure 1B) supported the interpretation that BEL and FKGK-11 targeted at least one common molecular entity. This might also indicate that alternate iPLA_2_ isoforms or proteoforms compensated for the loss, in the presence of BEL or FKGK-11. In contrast, using LY311727 to block sPLA_2_ resulted in a concentration-dependent decrease in endogenous FFA with no change in NBD-FFA, generated from NBD-PC labelled CSC (or NBD-PE labelled CSC; *n* = 1, not shown). These data indicated an active sPLA_2_ that could not access the exogenous NBD-PE or -PC (Figure 1A,B), despite the fact that NBD-PE was also seen to be a preferred substrate of CV luminal PLA_2_ (Figure 5A). Also, no significant change in the fluorescence intensity in the 30 min control, relative to the 0 min control (Figure 6C), indicated that 30 min was insufficient for any putative CV flippase to translocate PED6 (a modified PE) toward the CV lumen; had there been such translocation of PED6, a significant increase in fluorescence intensity should have occurred within 30 min due to luminal sPLA_2_ activity.

Notably, a significant increase in TAG in the BEL treated unlabelled CSC provided an early indication that iPLA_2_ could be the isozyme on CSC (Appendix A and [34,64]). The proposed ‘housekeeping’ function of iPLA_2_ is to generate lyso-phospholipid acceptors for the incorporation of FFA into membrane phospholipids [34,35,64,82]. Upon iPLA_2_ inhibition, lyso-phospholipid acceptor levels decrease, consequently leading to an increase in FFA concentration [29,64]. Under such a condition, the bulk of FFA is incorporated in the de novo metabolites TAG, DAG and MAG (Appendix A) [64]. BEL also inhibits magnesium-dependent phosphatidate phosphohydrolase (PAP), an enzyme catalysing the conversion of PA to DAG; however, there were no changes in PA in BEL treated CSC (Figure 1B).

Overall, the data suggested the presence of more than one catalytically active form of PLA_2_ on CSC, some of which may be strategically localized to maintain the docked and release-ready state of CV on the PM [29]. Blocking PLA_2_ activities using well characterized iPLA_2_ inhibitors (BEL [78] and FKGK-11 [35,67]) and an sPLA_2_ inhibitor (LY311727 [68]) caused significant inhibition of CSC fusion suggesting that even fully docked and fusion-ready CV are sensitive to changes in local PLA_2_ activity (Figure 1B and Figure 2).

### 5.2. Bioinformatics

With functional confirmation of the presence of active PLA_2_ species and association of their metabolites with the late steps of Ca^2+^-triggered exocytosis, we used a combined bioinformatics and top-down proteomics approach to identify and localize the active isoforms. Orthologs and paralogs are hypothesized to share high structural and functional similarity [83]; proteins essential to fundamental molecular mechanisms have a similar structure and function from urchin to mammals [37,73,74,83]. Accordingly, there was substantial homology in amino acid sequence between urchin and mammalian sPLA_2_ and iPLA_2_, particularly at the catalytic sites (Figure 3 and Table 1) [37]. The putative urchin sPLA_2_ shares conserved catalytic and regulatory regions of >75% identity and similarity with the mammalian orthologs (Table 1), including the highly conserved histidine and aspartic acid in the catalytic site consensus sequence (HDCCY) and the histidine in the Ca^2+^ binding loop (HCGVGG) (Figure 3A), at which LY311727 binds via its amide group [66]. Similarly, the iPLA_2_ share conserved catalytic and regulatory domains, including the conserved active site serine in the GXSXG consensus sequence that interacts with FKGK-11 [37,69] and the conserved cysteine residues are alkylated by the active BEL metabolites [78]. The urchin iPLA_2_ thus shares conserved catalytic and regulatory domains with the human and murine orthologs (Figure 3B and Table 1). Additionally, it also contains an ATP binding consensus motif (GGGVKG) and a N-terminal caspase-3 cleavage site (DVTD), that are all characteristic of iPLA_2_β (Figure 3) [35,84]. Notably, iPLA_2_β and sPLA_2_ genes are conserved in lower eukaryotes, further indicating an essential role of these enzymes [37].

### 5.3. Top-Down Proteomics Identify CV Associated PLA_2_ Isozymes

Based on the bioinformatics we then carried out targeted top-down proteomic analyses using 2DE and immunoblotting. These analyses identified a highly conserved catalytically active ~14 kDa CV luminal sPLA_2_ isozyme (Figure 4A) [25,26,36,85,86] having substrate preference for PE over PC (Figure 5A,B) [26]. Indeed, there are other reports of 14 kDa sPLA_2_ species having a substrate selectivity for PE over PC [17,87]. An increase in the activity of this CV luminal sPLA_2_ species in the presence of [Ca^2+^]_free_ was also confirmatory of its identification (Figure 5B) [25,27]. Similarly, using an antibody to the human form, we detected an ~63 kDa iPLA_2_ on both CSC and CV membranes (Figure 4B,C), although having different isoelectric points (pI). One of the most common post-translation modifications (PTM) causing a shift in pI is phosphorylation, resulting in an increase in the net negative charge on the protein. Hence, CV membrane iPLA_2_ isozymes with pI of ~5.2 and 5.4 may be phosphorylated (or otherwise modified) species relative to the bulk of the enzyme found on CSC (pI of ~6.2) (Figure 4B,C); the broad potential influence of protein phosphorylation on the late steps of exocytosis has been well documented [52]. The ~63kDa iPLA_2_ is proposed to arise from the proteolytic cleavage of a nascent full length 80–85 kDa iPLA_2_ [35,76,77]. Therefore, together with in-silico analysis (Figure 3B), identification of an ~63 kDa iPLA_2_ on the CV membrane (Figure 4C) and the significant decrease in endogenous FFA in BEL and FKGK-11 treated CV (Figure 8A,B) with a parallel increase in TAG (Appendix A), confirmed a catalytically active membrane bound CV iPLA_2_ [35,64,76,77,82]. Notably, on comparing CSC and CV data (Figure 4B,C), it would seem that much more of the iPLA_2_ may be localized to the PM, some (small) fraction presumably near or at sites of CV docking and fusion. The putative urchin iPLA_2_ contains 57 potential trypsin cleavage sites, predicted by PeptideMass [88] and thus cleavage at even a few of these sites would destroy the enzyme. Hence, loss of the 63 kDa CV iPLA_2_ following trypsin treatment (Figure 6B) and associated decrease in signal from the PLA_2_ selective PED6 substrate (Figure 6C), further confirmed the presence of an active iPLA_2_ on the external CV membrane. Thus, proteolytic removal of CV surface proteins, including iPLA_2_, caused a significant decrease in fusion extent to 54.3 ± 7.2% in the settle fusion assay that evaluates the priming and docking capacity of CV (Figure 6E), indicating that vesicle surface proteins including iPLA_2_ and SNAREs [29,41,42,43,44,45,89]―and possibly other as yet unidentified proteins―are critical for priming and/or docking. This is consistent with a suggested link between the CV surface PLA_2_ activity and influences on the late steps of the exocytotic pathway [29].

### 5.4. CV Associated PLA_2_ Isozymes Regulate Docking and/or Priming

Finally, we sought to test whether the identified PLA_2_ isozymes are components of the minimal molecular machinery present on CV which enables docking, Ca^2+^ sensing and membrane merger [38,42,43,45,47,48,49,50,51,52,55,90,91,92]. Thus, CV were treated with selective PLA_2_ inhibitors and changes in the fusion parameters were assessed using the well-established standard and settle fusion assays [38,42,43,45,47,48,49,50,51,52,53,54,90,91,92]. Consistent with the significant concentration-dependent decrease in fusion extent in CSC fusion assays, comparable effects were seen in the standard CV fusion assay following treatments with LY311727, BEL and FKGK-11 (Figure 7A). However, a far more pronounced decrease in the fusion extent in the settle assays (Figure 7B) confirmed that CV associated PLA_2_ isozymes or their metabolites play a critical role in docking and/or priming. The significant concentration-dependent decrease in endogenous CV FFA following the inhibitor treatments (Figure 8A,B), significant increase in PE (Table 2; in CV treated with 200 µM LY311727) and parallel increase in TAG (Appendix A; in CV treated with 500 µM BEL) confirmed that these inhibitors targeted CV associated PLA_2_ isozymes.

Furthermore, iPLA_2_ hydrolyse BEL to generate diffusible Bromomethyl Keto Acid (BMKA), which alkylates cysteine thiols in iPLA_2_ causing its inhibition, in addition to non-specific alkylation of cysteine thiols present in other proteins [78]. Therefore, the significant reduction in the Ca^2+^ sensitivity of CSC treated with 500 µM BEL (Figure 2), relative to a lack of effect on CV treated identically (Figure 7A,B), suggested that BMKA acted (to some extent) like N-ethylmaleimide (NEM) in CSC [90,92]. This indicates additional thiol sites on the PM that regulate efficiency of the fusion mechanism (i.e., Ca^2+^ sensing). In terms of targeted inhibition, it should be noted that the compounds, LY311727, BMKA and FKGK-11 are highly hydrophobic with log P values of 1.92 ± 0.62, 3.90 ± 0.37 and 5.40 ± 0.89, respectively, where the partition coefficient (P) = [organic]/[aqueous] (calculated using ADC/ChemSketch [93]). Hence, these compounds have a high likelihood to intercalate into or cross the CV membrane. Indeed, the significant concentration-dependent decrease in fusion extent upon 200 µM LY311727 treatments, to 33.6 ± 20.8%, 80.0 ± 10.7% and 53.5 ± 13.4% in the CSC, CV-CV standard and settle fusion assays, respectively, (Figure 2A,B and Figure 7A,B) and; the parallel decrease in endogenous FFA (Figure 1A and Figure 8A,B) indicated that inhibition of the CV luminal sPLA_2_ reduced CV priming and/or docking. Although BMKA and FKGK-11 may also have crossed the CV membrane, it seems unlikely that these would inhibit the sPLA_2_―its sulfhydryl groups would already be engaged in intra-molecular disulphide bonding and thus would not be susceptible to BMKA induced alkylation [78] and the absence of a catalytic site serine would obviate FKGK-11 binding [67]. Therefore, overall, the inhibitors chosen are expected to be highly selective for the PLA_2_ species in question and thus reflect selective influences on the functional assays.

### 5.5. Concluding Remarks

According to the Stalk-pore hypothesis, the fusion of two distinct membrane bilayers proceeds through sequential steps resulting in the localized mixing of proximal monolayers followed by fusion of the distal monolayers [94,95,96,97]. This enables vesicle content mixing (homotypic fusion) or release (heterotypic fusion) and subsequent content dispersal. PLA_2_ activity produces both negative and positive curvature lipids—LPC and FFA, respectively—which are known to inhibit or promote fusion depending on their site of incorporation [10,11,29,94,95,98]. In this regard, localization of the endogenous catalytically active PLA_2_ isozymes that supply LPC and FFA near the docking and/or fusion site is of critical importance. The internal CV microenvironment has a pH of ~5.5 and contains ~100 mM calcium, mostly in a bound state, as has also been found in other (mammalian) secretory vesicles [30,86,99,100,101]; however, the estimated global luminal [Ca^2+^]_free_ is ~1–10 µM, occurring as transients, linked in part to the opening of p-type Ca^2+^ channels [30]. As these are global measures, the local [Ca^2+^]_free_ at the CV membrane would be much higher. This is also consistent with the estimates of luminal [Ca^2+^]_free_ in chromaffin granules under basal conditions [100]. Hence, in response to CV luminal Ca^2+^ transients in vivo, the activity of localized luminal sPLA_2_ would increase, preferentially hydrolysing PE on the inner CV monolayer (Figure 4A and Figure 5A,B ), at or immediately adjacent to the docking/fusion site. Thus, under basal conditions, the external CV membrane associated iPLA_2_ generates LPC and FFA by cleaving PC on the outer CV monolayer, while luminal sPLA_2_ produces LPE and FFA on the inner CV monolayer. Such localization of iPLA_2_ on the outer vesicle (and inner plasma) membrane, near docking/fusion sites, would block spontaneous merger of the vesicle and PM due to high local levels of LPC at membrane contact sites, in particular considering the substantially higher positive intrinsic curvature of LPC relative to LPE [98]. As such, this might act as a native fusion ‘brake’ that must be overcome by the Ca^2+^-triggered fusion mechanism [29], in which basal LPC production is somewhat balanced by intermittent LPE production via transient sPLA_2_ activity (Figure 9). In this regard and that of FFA (see below) it would also be important to assess the distribution of fatty acid species at sn-1 and sn-2 of the endogenous CV membrane PC and PE. The net accumulation of FFA on *cis* or *trans* monolayers and any associated effects would be dependent on the rate of passive diffusion, flipping efficiency, rate of metabolism, site of incorporation, chain length and degree of unsaturation [10,95,102,103,104,105,106,107]. Notably, perhaps complimentary to the actions of LPC and LPE, PLA_2_ derived FFA has been proposed to support the formation of *trans* SNARE complexes to maintain the primed and/or docked state of vesicles [4,29,108,109]. Importantly, basal CV luminal sPLA_2_ activity supported CV-PM and CV-CV fusion (Figure 2, Figure 6D,E and Figure 7A,B); whether the activity is also sensitive to transient changes in luminal [Ca^2+^]_free_ and how these may be linked to the final membrane merger steps are unknown. However, the likely coordinated action (i.e., cross-talk) of the CV luminal sPLA_2_ and outer membrane bound iPLA_2_ in maintaining the docked, fusion-ready state of vesicles on the PM is evident as both interventions used here―pharmacological inhibition of CV luminal sPLA_2_ and membrane associated iPLA_2_ and proteolytic removal of iPLA_2_―significantly impaired priming and/or docking.

## 6. Conclusions

The data confirm (i) the presence of endogenous, catalytically active CV associated PLA_2_ isozymes―a CV luminal sPLA_2_ and an iPLA_2_ associated with the outer CV membrane; and (ii) that proteolytic removal of CV surface proteins including iPLA_2_ or the pharmacological inhibition of both the isozymes significantly reduces priming and/or docking. Based on this as well as previously measured vesicular Ca^2+^ transients, basal Ca^2+^ dependent CV luminal sPLA_2_ activity and CV membrane attached iPLA_2_ would appear to act together to support efficient vesicle priming and/or docking. While intriguing, this warrants further study, particularly relating to the apparent effects of FFA on the formation of SNARE complexes and/or in modulating membrane biophysical properties, as FFA is also known to induce the transition from the bilayer to the hexagonal phase upon its incorporation in the membrane [10,107,110].

## Figures and Tables

**Figure 1 cells-08-00303-f001:**
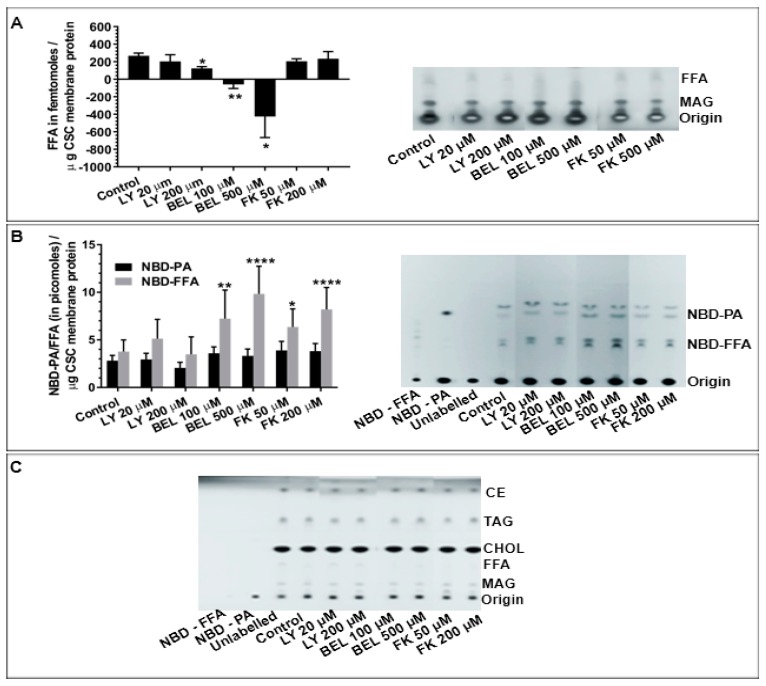
Detection and quantification of (**A**) endogenous CSC FFA, *n* = 3 separate experiments; (**B**) NBD-FFA and -PA in intact CSC after 15 min labelling with NBD-PC followed by 20 min inhibitor treatments, *n* = 3 separate experiments; (**C**) total CSC neutral lipids visualized on the same TLC plate as shown in B, after charring with CuSO_4_. This also confirmed consistent lipid loading in each lane. Representative chromatograms showing indicated changes in lipids. (*p*-value, * < 0.05, ** < 0.005, **** < 0.0001 indicates relative difference to the control). Note: A section of chromatogram is shown in A and lanes in all chromatograms are grouped together from the same HPTLC plate following removal of lanes not associated with this study. A standard neutral lipid protocol was used to resolve the lipids [56].

**Figure 2 cells-08-00303-f002:**
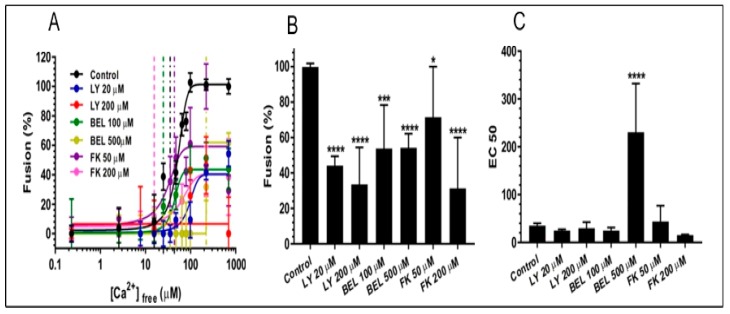
Effects of LY311727 (selective for sPLA_2_) and BEL and FKGK-11 (both selective for iPLA_2_) on (**A**) CSC Ca^2+^ activity curves (i.e., exocytosis *in vitro*); (**B**) fusion extent and; (**C**) Ca^2+^ sensitivity (EC50); *n* = 3–14. (*p*-value, * < 0.05, *** < 0.0005, **** < 0.0001 indicates difference relative to the control).

**Figure 3 cells-08-00303-f003:**
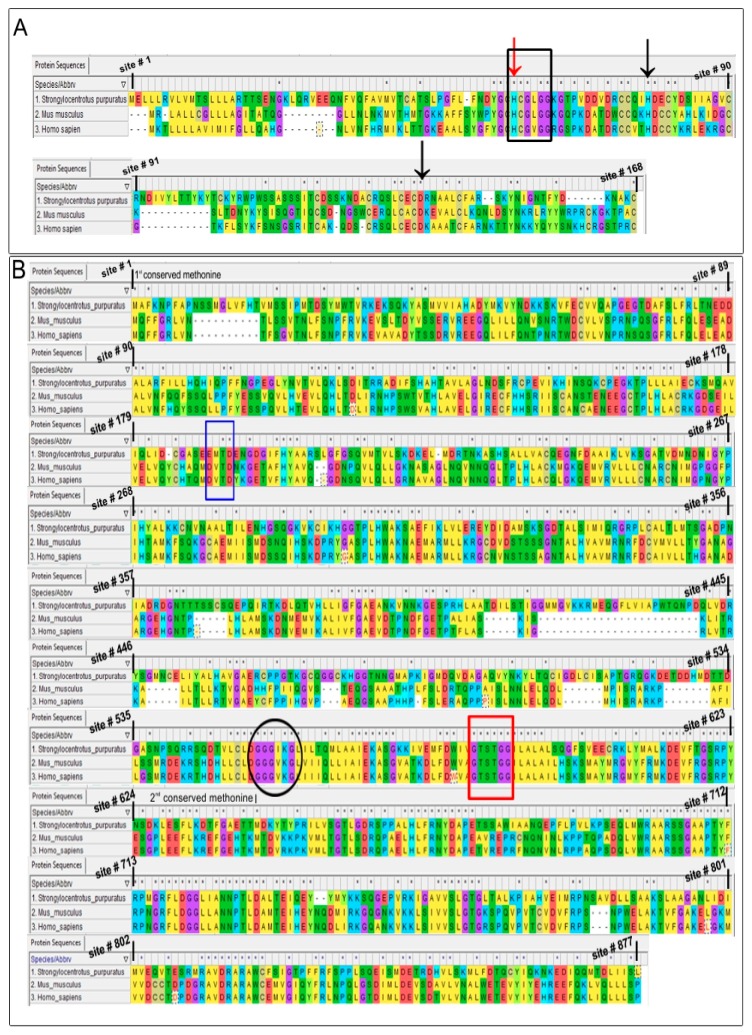
Sequence alignment of predicted urchin (**A**) sPLA_2_ and (**B**) 80–85 kDa iPLA_2_. The first row in each panel is the amino acid sequence of urchin PLA_2_ isozymes and the subsequent two rows are of murine and human PLA_2_ isozymes, respectively. Conserved amino acids are indicated with an asterisk and site # represents the amino acid residue number in the primary urchin PLA_2_ sequence (also considering gaps). The red arrow indicates conserved Histidine in the Ca^2+^ binding loop (Black box), the site at which LY311727 binds. Black arrows show conserved Histidine and Aspartic acid which form the catalytic site His-Asp dyad. Blue and red boxes indicate the caspase cleavage site and the active site serine in the GXSXG consensus sequence, respectively. The circle indicates the nucleotide binding site. Dissimilar residues with the same background colour indicate conservative substitution.

**Figure 4 cells-08-00303-f004:**
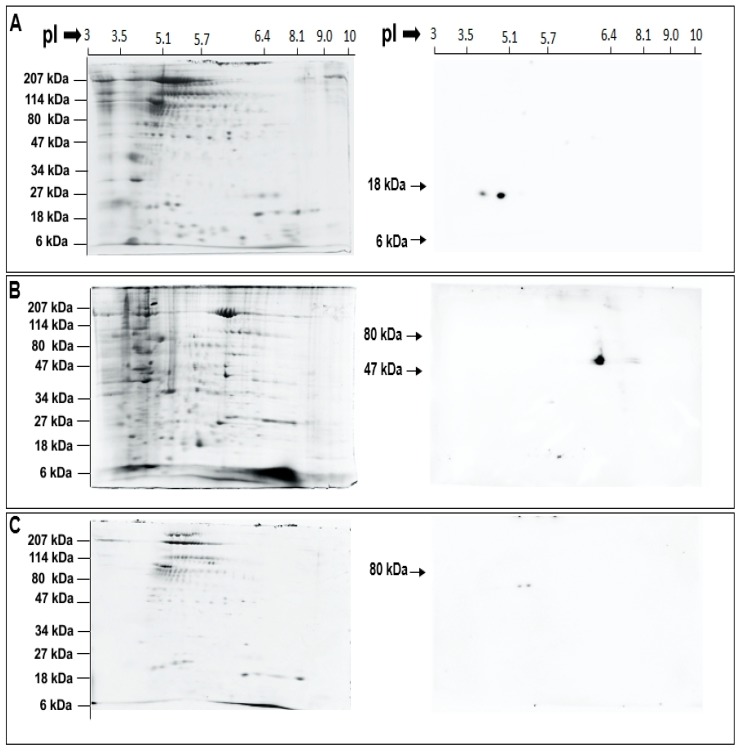
Immuno-reactive spots detected by (**A**) sPLA_2_ antibody in the CV luminal fraction; (**B**) iPLA_2_ antibody in the CSC membrane fraction; and (**C**) iPLA_2_ antibody in the CV membrane fraction. cCBB stained 2D SDS-PAGE gels (left) and parallel western blots (right).

**Figure 5 cells-08-00303-f005:**
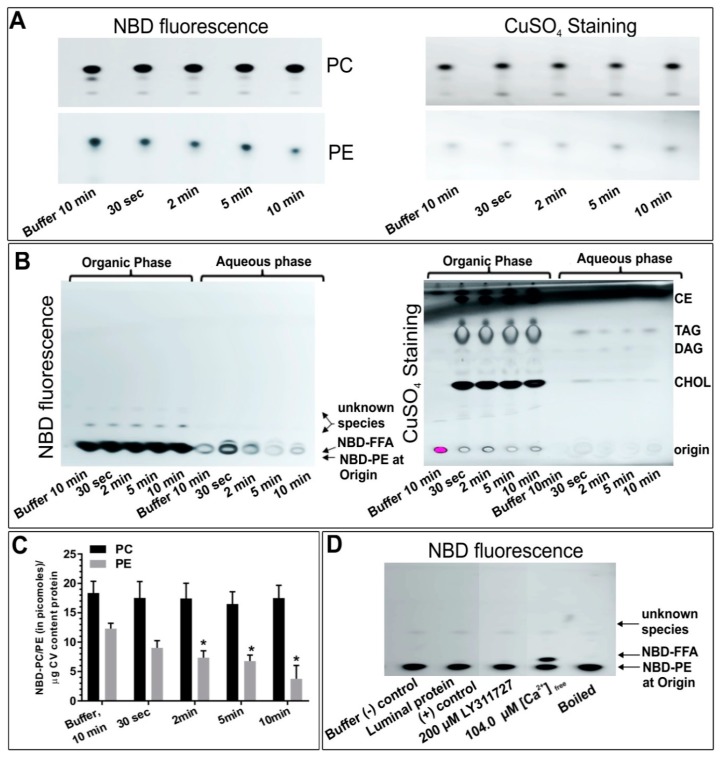
(**A**) Time-dependent decrease in NBD-PE and no change in NBD-PC, when incubated with isolated CV luminal proteins, *n* = 3; (**B**) Detection of NBD-FFA generated from NBD-PE, when incubated with CV luminal proteins; *n* = 2–3. After incubations, activity was stopped by adding 0.2 µL HCl, total lipids isolated, resulting organic and aqueous phases retained, dried and resolved. (**C**) Bar graph summarizing the changes in NBD-PC and -PE, indicating the presence of CV luminal PLA_2_ with high catalytic activity for PE. (**D**) Effect of LY311727, increased [Ca^2+^]_free_ and boiling on luminal PLA_2_ activity assessed by incubating isolated CV luminal proteins with NBD-PE; *n* = 3. (*p*-value, * < 0.05 indicates difference relative to the control). The chromatograms in A were resolved using a standard phospholipid protocol and in (**B**,**D**) using an established protocol for neutral lipids [56]. Note: Left panels in (**A**,**B**) show NBD fluorescence and the right panels show same section of the chromatogram after charring with CuSO_4_ [56]. The lanes shown in (**D**) are grouped together from the same chromatogram following removal of lanes not associated with this study.

**Figure 6 cells-08-00303-f006:**
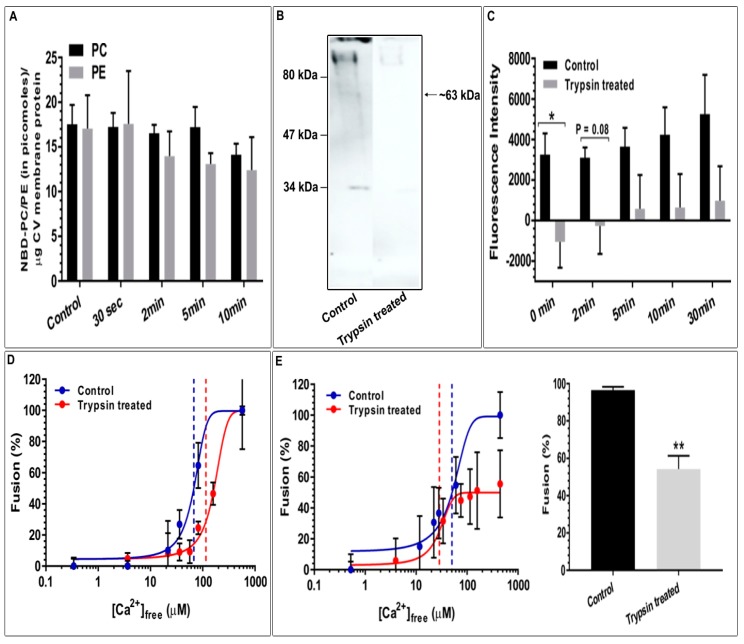
(**A**) Time-dependent changes in NBD-PC and -PE, when incubated with CV membrane fragments (*n* = 2–3); (**B**) Western blot showing an apparent complete loss of iPLA_2_ in trypsin treated CV (*n* = 3). A total of 5µg CV membrane protein per lane was resolved; (**C**) Time-dependent changes in the CV surface PLA_2_ activity in the intact CV assessed using PLA_2_-selective substrate PED6 [79,80] (*n* = 3). The loss of the fluorescence signal is consistent with the loss of PLA_2_ from the CV surface following ablation with trypsin. Functional assays showing the effects of trypsin treatment on CV-CV Ca^2+^ activity curves [45]; (**D**) Standard fusion assay (*n* = 2); (**E**) CV settle fusion assay (*n* = 3). (*p*-value, * < 0.05, ** < 0.005 indicates difference relative to the control). Note: The lanes in (**B**) are grouped together from the same western blot following removal of intervening lanes.

**Figure 7 cells-08-00303-f007:**
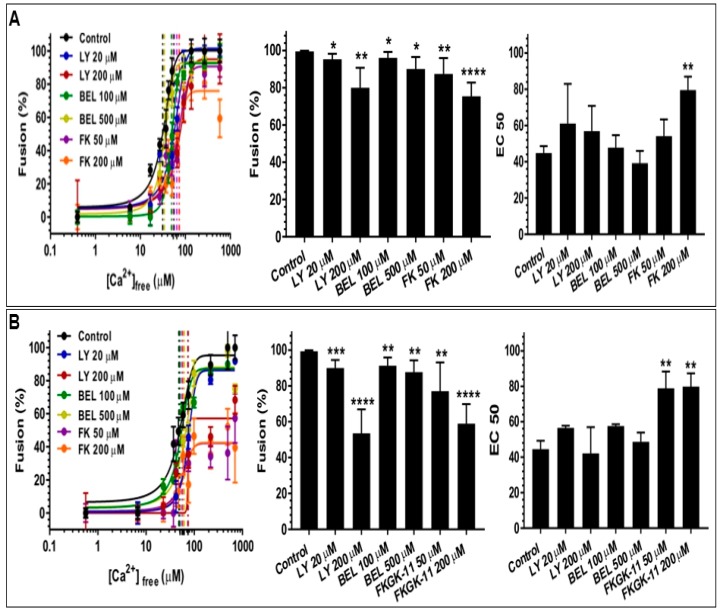
Effects of LY311727 (selective for sPLA_2_) and BEL and FKGK-11 (both selective for iPLA_2_) on CV-CV Ca^2+^ activity curves, fusion extent and Ca^2+^ sensitivity (EC50); (**A**) standard fusion assay, *n* = 3–19; (**B**) settle fusion assay, *n* = 3–19. (*p*-value, * < 0.05, ** < 0.005, *** < 0.0005, **** < 0.0001 indicates difference relative to the control.

**Figure 8 cells-08-00303-f008:**
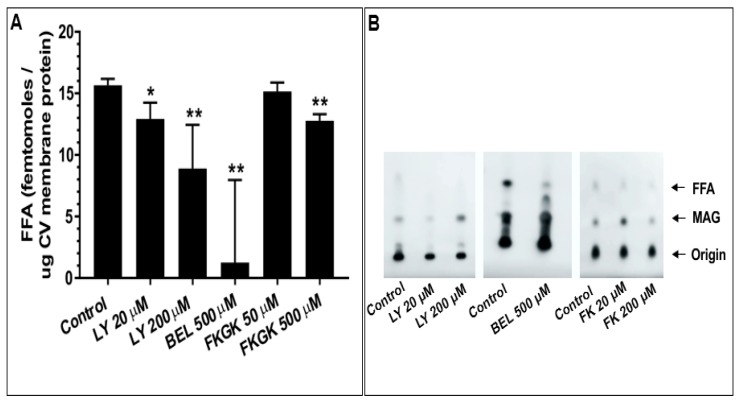
(**A**) Changes in the endogenous FFA in CV treated with indicated doses of LY311727, BEL and FKGK-11; *n* = 4–12; (**B**) Section of the chromatograms showing the changes in endogenous FFA. (*p*-value, * < 0.05, ** < 0.005 indicates difference relative to the control). A standard neutral lipid protocol was used to resolve the lipids [56].

**Figure 9 cells-08-00303-f009:**
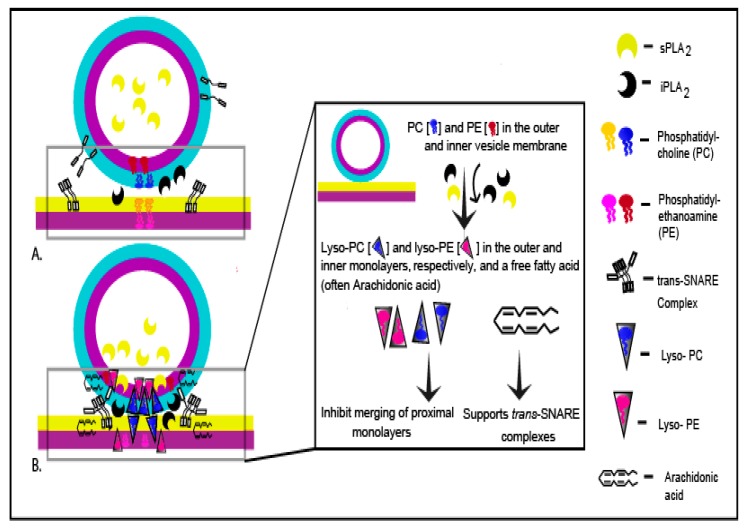
Working hypothesis: (**A**) Non-contacting membranes (**B**) Vesicle docked with the plasma membrane (PM). At basal rate, the vesicle luminal sPLA_2_ and membrane bound iPLA_2_ cleave PE and PC in the inner and outer monolayer, respectively; PM bound iPLA_2_ also cleaves PC in the inner monolayer of the PM. This causes a local increase in LPC at the membrane contact site that is balanced by intermittent LPE production at the adjacent inner membrane via transient sPLA_2_ activity. This mechanism may thus act as a native fusion ‘brake’ and the released FFA (often Arachidonic acid) may support *trans* SNARE complex formation to ensure and maintain efficient priming/docking. (adapted from [29] with permission).

**Table 1 cells-08-00303-t001:** Identity and similarity of urchin PLA_2_ isozymes to the human and murine forms.

	Target Species	Percent Identities	Percent Similarities	Percent Identities	Percent Similarities
**Urchin Isozymes**		**(Full Length)**		**(at Histidine Catalytic Site)**	
**sPLA_2_**	Human	39	50	68	67
	Murine	36	46	74	74
		**(Full Length)**		**(at Serine Catalytic Site)**	
**iPLA_2_**	Human	36	52	100	100
	Murine	36	53	100	100

**Table 2 cells-08-00303-t002:** Endogenous PE and PC levels in (**A**) CSC and (**B**) CV, treated with the indicated doses of the inhibitors. ND-not done; NA-not applicable.

A											
PE (Picomoles/µg CSC Membrane Protein)		Mean	SEM	Student *t*-Test	Sample Size (*n*)	PC (Picomoles/µg CSC Membrane Protein)		Mean	SEM	Student *t*-Test	Sample Size (*n*)
**Control**		29.5	7.5		5	**Control**		47.7	11.0		6
**LY311727**	**20 µM**	ND	NA	NA	NA	**LY311727**	**20 µM**	66.6	3.0	0.38	2
**LY311727**	**200 µM**	ND	NA	NA	NA	**LY311727**	**200 µM**	38.9	2.2	0.68	2
**BEL**	**100 µM**	ND	NA	NA	NA	**BEL**	**100 µM**	ND	ND	NA	NA
**BEL**	**500 µM**	28.0	7.5	0.90	3	**BEL**	**500 µM**	77.2	5.0	0.31	3
**FKGK-11**	**50 µM**	51.0	7.6	0.11	3	**FKGK-11**	**50 µM**	63.2	19.1	0.47	3
**FKGK-11**	**200 µM**	31.8	9.3	0.85	3	**FKGK-11**	**200 µM**	50.7	8.9	0.86	3
**B**											
**PE (Picomoles/µg CV Membrane Protein)**		**Mean**	**SEM**	**Student *t*-Test**	**Sample Size (*n*)**	**PC (Picomoles/µg CV Membrane Protein)**		**Mean**	**SEM**	**Student *t*-Test**	**Sample Size (*n*)**
**Control**		46.0	5.5		11	**Control**		75.1	11.8		13
**LY311727**	**20 µM**	ND	NA	NA	NA	**LY311727**	**20 µM**	96.4	26.3	0.5	3
**LY311727**	**200 µM**	70.5	5.1	0.05	3	**LY311727**	**200 µM**	114.2	19.9	0.1	6
**BEL**	**100 µM**	ND	NA	NA	1	**BEL**	**100 µM**	ND	NA	NA	NA
**BEL**	**500 µM**	46.6	12.7	0.96	5	**BEL**	**500 µM**	57.3	7.2	0.4	5
**FKGK-11**	**50 µM**	40.4	12.9	0.66	3	**FKGK-11**	**50 µM**	53.5	26.0	0.4	3
**FKGK-11**	**200 µM**	22.5	6.7	0.06	3	**FKGK-11**	**200 µM**	42.4	16.5	0.2	3

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
