# Peer review of "Combined Targeted Omic and Functional Assays Identify Phospholipases A2 that Regulate Docking/Priming in Calcium-Triggered Exocytosis"

_cells, 2019, doi:10.3390/cells8040303_

Round 1

Reviewer 1 Report

Phospholipase A2 is a superfamily of the sixteen PLA2 enzyme families known to catalyze the hydrolysis of the sn-2 ester bond in phosphatidylcholine (PC) and/or phosphatidylethanolamine (PE), producing a lysophospholipid and a fatty acid. Both may function as intracellular second messengers and serve as precursors for other bioactive lipids (Balsinde et al., 1999; Liu et al., 2016; Murakami et al., 1997; Shayman, 2016). PLA2 has been suggested to play a role in vesicle fusion in many biological systems (Nakamura, 1993; Brown et al. 2003; Juhl et al., 2003), and lysolipids produced by PLA2 activities have been shown to inhibit membrane fusion (Chernomordik et al., 1995). It has also been demonstrated that lysolipids regulate line tension of boundaries of lipid rafts (Lyushnyak et al., 2018), which are known as Ca2+-triggered membrane fusion sites (Gil et al., 2005; Churchward et al., 2005). Thus, the identification of specific PLA2 isozymes present in the vesicle lumen/membrane as well as understanding their precise functions in late stages of triggered release is important.

This study utilizes sea urchin egg preparations (such as cell surface complexes (CSC) and fully-docked, primed, and release-ready cortical vesicles (CV)).  This preparation allows for the direct assessment of the individual steps in exocytosis and for quantitative dissection of the molecular mechanism underlying late steps of regulated exocytosis (eg. docking, priming, and fusion per se) at both functional and molecular levels. The possibility to isolate and study these stage-specific preparations is clearly an important advantage and strength of this model system.

The study demonstrates PLA2 association with cortical vesicles and distinguishes between Ca2+ independent iPLA2 working within the outer CV membrane and sPLA2 functioning in the CV lumen. The authors demonstrate that suppressing activities of those PLA2 enzymes with selective inhibitors affects priming and/or docking, which implies that a combined contribution of both isozymes is critical for the efficiency of fusion. The manuscript contains important information of interest for other investigators and thus makes a relevant contribution to the field.I recommend the manuscript for publication after addressing the following comments.

To improve the clarity of the messages, the authors should clarify or answer the following points to strengthen the manuscript:

1.   Although well presented, another round of copy-editing would be welcome. Certain words seem to be missing (eg. line 194: “essentially ….” may be missing a word?), and various sentences could be shortened or rewritten.

2.   The figure legends are very brief, do not clearly describe the meaning of the results, and miss critical information. This makes the figures difficult to understand. For instance, Fig.5A shows organic and aqueous lipid phases with no description in the figure’s legend. Readers should be able to have a general understanding of figures without having to refer to the text, so it would be beneficial to have more descriptive legends.

3.   “Dose-dependent” should be changed to “concentration-dependent” as dose dependence relates more to oral drug administration.

4.   Fig. 1B. The figure shows separation of both NBD-FFA and NBD-PA.   It would be useful if the authors could clarify which protocol was used to separate those lipids.  In the method section, the authors indicate that they used the neutral lipid protocol for neutral lipid separation (eg. TAG, cholesterol, and FFA), while phospholipid separation protocol was used to separate phospholipids (eg. PE, PC, PA). Then it becomes unclear how both NBD-FFA and NBD-PA were resolved on the same plate. FFAs are normally separated along with neutral lipids, while PA is resolved by using a phospholipid protocol. Also, the surface of silica gel is highly polar; thus, polar lipids in the solvent would interact strongly with the surface of the silica gel and move slowly. Lipids that are more hydrophobic interact weaker and move quicker. Thus, FFA should move to the top of the plate while PA is more polar and should be closer to the bottom of the plate if separated together. Fig.1B shows the opposite.  This needs to be explained/addressed in the manuscript. 

5.   Fig.1C shows resolved neutral lipids. No DAG is present there. Is it bellow detection levels? If the same plate was used for CuSO4 charring (Fig.1C) and NBD fluorescence (Fig.1B), then it becomes unclear how polar PA was detected on the same plate (Note: only neutral lipids are shown on Fig.1C and there is no PA there). The Fig.1 legend says that CSC were resolved for lipids after labeling with NBD-PC and inhibitor treatments.  It would help if PC levels after CuSO4 staining would be shown for tested conditions to assess changes.  It seems logical to include a representative image of resolved phospholipids in Fig.1C for quantification of PC and/or  PE levels.  The authors need to check the data and/or give more details on the protocol used.

6.   Fig 5A,B.  Again FFA and phospholipids (PE) are shown on the same plate. It would help if the authors could provide details of the protocol used to allowed PE (very polar) and FFA (much more hydrophobic) to be side by side on the plate. Further, it would be very informative if the authors would be able to demonstrate resolved lipids after CuSO4 charring on the same HPTLC plate.

7.   Line 500-501. In the text the authors refer to Supplementary Fig. 1A, B concerning DAG levels. However, no DAG was detected/shown on that figure.

8.   Line 489. The authors claim that luminal sPLA can not access exogenous NBD-PE and NBD-PC. However, NBD phospholipids are known to quickly flip-flop across the membrane (matter of minutes; Stevens et al., 2008; Sharom, 2011). It would be informative if the authors address this point.

Reviewer 2 Report

The work by Dabral and Coorssen identifies the expression and localization of phospholipase A2 isoforms (PLA2) in cortical vesicles of the sea urchin and subsequently attempts to examine their functional role in the fusion of these cortical vesicles. The authors use a combination of methodologies to determine whether arachidonic acid production in preparation of CVs or CVs with the plasma membrane is modified by PLAinhibitors. In parallel, they determine the efficiency of fusion under the exposure to the same inhibitors. The authors address a fundamental question in the field of vesicle exocytosis – are there short-term changes at the lipid composition of vesicles and in the acceptor plasma membrane that support exocytosis (and possibly also endocytosis)?. While the identification of the PLAisoforms based on bioinformatic and biochemical measures is rather convincing, the experiment testing for the functional relationship between PLAactivity and fusion through the use of inhibitors require some further clarifications. I therefore support the publication of this manuscript in ‘Cells’ after revision.

Introduction:

-       The title of the paper suggests that ‘omics’ methodologies are used to reach the paper’s conclusion. However, the methodologies used in the paper largely fall under the definition of biochemical methods. Please reconsider revising the title.

-       the authors indicate that the expression of PLAhas been documented mostly in cells where large-dense core vesicles are the primary vesicle type to be released, and indicate that in older work, the expression of PLAwas suggested also in neurons where small, clear, synaptic vesicles (SVs) are primarily released. The authors cite papers where a proteomic analysis of SVs has been carried out to point out that recent studies do not seem to support this notion. Can PLA2isoforms found at neuronal synapses at all? Looking into studies that explored the proteomic composition of the synapse may be of interest to the reader.

-       Line 68 ‘C-2’ should be ‘C2’

-       iPLA2is later identified as a protein of the CV fraction. How do iPLAisoforms interact with the membrane? Could the authors speculate why iPLA2 is isolated with CVs? 

Results:

-       At the end of the first paragraph, the authors indicate that BEL and FKGK-11 are iPLA2inhibitors, whereas LY is an sPLAinhibitor. In Figure 1A, 200µM LY, and BEL, lead to a reduction in FFA levels, while FK has no effect, indicating a blockade of a putative PLAactivity. Since the identity of PLAisoforms, and especially their drug sensitivity in sea urchin is unknown, it is conceivable that the protein variants will be sensitive to one, but not to the other inhibitor. However, in figure 1b, in which, to my understanding, the detection method only is changed, BEL and FK both now have an effect, of enhancing PLAactivity? If another ‘BEL insensitive’ isoform exists, why is its function unraveled only when BEL is present? Shouldn’t the levels be comparable to that of WT?  Please further clarify the discrepancy between figure 1A and 1B.

-       In Figure 2 the authors show a large effect of all inhibitors on fusion. How does this agree with the results in Figure 1A, where FK does not exert any effect? 

-       Figure 4: The authors are using human antibodies to identify sea urchin proteins. To univocally identify PLAin the 2-D gels, a mass spectrometric analysis of the regions where the western blot analysis indicated a PLAspot is desired. A mass-spectrometric detection, in this case, would make the conclusion of the paper significantly stronger. Can the authors comment on the feasibility of such an experiment?

-       Figure 5, legends – the comment regarding the removal of lanes in the chromatogram should refer to figure 5B, not 4B.

-       It is highly unclear, how the experiment presented in figure 1A and Figure 2 are different than the one in Figure 8 and figure 7, as the authors themselves state that CV-PM fusion is indistinguishable than CV-CV fusion. In fact, CV-PM fusion may be considered by many as ‘more physiological’. How does the experiment in Figure 7+8 provide the reader with additional/complementary information to that of Figure 1A+2? please elaborate and consider grouping the two together.

-       Regarding Figure 6 and the effect of iPLA2– How do the authors settle the controversy between figure 6A (no effect) and 6C? How do the results in figure 6A can be settled with the large effects of BEL and FK on fusion (Figure 2, 7) and with the large effect on FFA production (Figure 1B, 8)? Why is PED6 suddenly being used as substrate? And how do vesicles with a trypsinized coat still maintain 50%-100% Ca2+-dependent fusion ability?

-       Line 555 ‘.’ is missing after ‘(Fig. 3B)’.

In summary, throughout the paper, the data collected as evidence for the sPLAexpression and effect are rather consistent and convincing – from the effect of LY on FFA production and fusion, to the strong signal at the 2-D gel, and to the results presented in figure 5, using a luminal CV fraction. In contrast, the results regarding the expression of iPLAare less convincing, and should be further clarified.

Author Response

We thank the reviewers for their interest in and support of our work. We appreciate the opportunity to respond to their comments and questions, believe that we have now effectively clarified matters raised, and hope that the manuscript is now acceptable for publication.

Reviewer-2

Comments and Suggestions for Authors

The work by Dabral and Coorssen identifies the expression and localization of phospholipase A2 isoforms (PLA2) in cortical vesicles of the sea urchin and subsequently attempts to examine their functional role in the fusion of these cortical vesicles. The authors use a combination of methodologies to determine whether arachidonic acid production in preparation of CVs or CVs with the plasma membrane is modified by PLAinhibitors. In parallel, they determine the efficiency of fusion under the exposure to the same inhibitors. The authors address a fundamental question in the field of vesicle exocytosis – are there short-term changes at the lipid composition of vesicles and in the acceptor plasma membrane that support exocytosis (and possibly also endocytosis)?. While the identification of the PLAisoforms based on bioinformatic and biochemical measures is rather convincing, the experiment testing for the functional relationship between PLAactivity and fusion through the use of inhibitors require some further clarifications. I therefore support the publication of this manuscript in ‘Cells’ after revision. 

Introduction

1.                  The title of the paper suggests that ‘omics’ methodologies are used to reach the paper’s conclusion. However, the methodologies used in the paper largely fall under the definition of biochemical methods. Please reconsider revising the title.

Response: While we respectfully note the reviewer’s thoughts, we do not agree. The methodological foundations of this work are well-established Top-down Proteomic, Lipidomic (i.e. ‘Omic) as well as functional assays. We thus feel the title is quite appropriate. If the reviewer feels strongly about this, we would appreciate being provided with a reference or other such source which specifically notes that “…the methodologies used in the paper largely fall under the definition of biochemical methods.” Coupled with the bioinformatics, the protein and lipid analyses are well-established top-down ‘omic’ approaches that the senior author is recognized for and has indeed published extensively on, rather than being shotgun bottom-up analyses. If this is what the reviewer is concerned about, we could add ‘top-down’ omic analyses to the title? Again, respectfully, we are happy to discuss this further if the reviewer feels particularly strongly about it, but then again, this is a special issue on the application of proteomics in cell biology research; the senior author had assumed that was a key reason he was invited to contribute to the special issue.

2.                  the authors indicate that the expression of PLAhas been documented mostly in cells where large dense core vesicles are the primary vesicle type to be released, and indicate that in older work, the expression of PLAwas suggested also in neurons where small, clear, synaptic vesicles (SVs) are primarily released. The authors cite papers where a proteomic analysis of SVs has been carried out to point out that recent studies do not seem to support this notion. Can PLA2isoforms found at neuronal synapses at all? Looking into studies that explored the proteomic composition of the synapse may be of interest to the reader.

Response: Thank you for the suggestion.

Recently, Seok et al., 2016 in “Identification of long-lived synaptic proteins by proteomic analysis of synaptosome protein turnover” identified iPLA2 gamma in synaptosomes. Ken et al., 2016 in “Proteomic Analysis of Unbounded Cellular Compartments: Synaptic Clefts” identified sPLA2 and iPLA2 isoforms in the synaptic cleft. However, the latter were considered to be non-synaptic cell surface proteins. Thus, the Seok et al reference has been added to the manuscript and the relevant sentences regarding identification of synaptic / SV PLA2 isoforms slightly reworked for clarity.

3.                  Line 68 ‘C-2’ should be ‘C2’

Response: Thank you, C-2 has been changed to C2.

4.                  iPLA2 is later identified as a protein of the CV fraction. How do iPLAisoforms interact with the membrane? Could the authors speculate why iPLA2 is isolated with CVs?

Response: The iPLA2 is a membrane associated protein, that interacts with membranes via its interfacial binding region that forms hydrophobic, ionic and hydrogen bonds (Mouchlis et al. 2015 PNAS). According to MD simulation, region 708-730 forms an amphipathic helix that penetrates into the membrane, thus acting as an anchor (Mouchlis et al. 2015 PNAS, Mouchlis et al. 2015 Adv Biol Regul). It is likely due to these strong molecular interactions that iPLA2 remained attached to the CV membrane during isolation and thus was detected in the membrane fraction by western blotting; we know from extensive previous proteomic analyses over almost 20 years that, regardless of the tissue being analyzed, the simple lysis and ultracentrifugation protocol used here to recover a membrane pellet results in the isolation of both integral and membrane-associated proteins. iPLA2 is also known to interact with Ras-related C3 botulinum toxin substrate 1, Protein kinase C - α, ε (PKC - α, ε), phospholipase D (PLD) and Voltage-dependent anion-selective channel protein 1 (VDAC1), most of these are on the CV membrane and could thus also serve as anchors or localizers for iPLA2. Clearly then, considering the data presented, iPLA2 is isolated with CV as it has a critical role in the late steps of the exocytotic pathway. 

Results

1.                At the end of the first paragraph, the authors indicate that BEL and FKGK-11 are iPLA2 inhibitors, whereas LY is an sPLAinhibitor. In Figure 1A, 200µM LY, and BEL, lead to a reduction in FFA levels, while FK has no effect, indicating a blockade of a putative PLAactivity. Since the identity of PLAisoforms, and especially their drug sensitivity in sea urchin is unknown, it is conceivable that the protein variants will be sensitive to one, but not to the other inhibitor. However, in figure 1b, in which, to my understanding, the detection method only is changed, BEL and FK both now have an effect, of enhancing PLAactivity? If another ‘BEL insensitive’ isoform exists, why is its function unraveled only when BEL is present? Shouldn’t the levels be comparable to that of WT?  Please further clarify the discrepancy between figure 1A and 1B.

Response: Fig 1A show changes in endogenous FFA upon inhibitor treatment and Fig 1B the changes in NBD-FFA and -PA that were generated from exogenous NBD-PC incorporated in the CSC membranes. As NBD-PC is a common substrate for all PLA2 isozymes, blocking ‘BEL sensitive’ species (which also seemed to interact at least to some extent with FKGK-11), likely resulted in compensatory activity by a ‘BEL insensitive’ species. Thus, NBD-FFA levels increased rather than returning to control levels. This would only occur if isolated CSC carries multiple PLA2 isozymes; this seems quite likely considering the vast amount of PM that makes up CSC (see also answer below as to why we work with CV). As NBD-PC would have randomly incorporated into all the membranes present in CSC (i.e. including attached subplasmalemmal ER), any phospholipases present would be capable of utilizing it if it is an appropriate substrate (i.e. note also PLD activity in the preparation; PLC activity has been previously documented as well). Clearly, in native (i.e. unlabelled) CSC (Fig. 1A), there is a tighter regulation of substrate availability and use.

2.                In Figure 2 the authors show a large effect of all inhibitors on fusion. How does this agree with the results in Figure 1A, where FK does not exert any effect? 

Response: The basal level of endogenous FFA in CSC is ~260 femtomoles / µg CSC membrane protein, relative to ~15 femtomoles / µg CV membrane protein (figure 1A and 8); noting again that CSC is a mix of membrane types. In isolated CV, lower and higher doses of FKGK-11 reduced endogenous FFA by ~ 6% and ~ 13%, respectively (Fig. 8). This suggests that a small % change occurred in endogenous FFA in FKGK-11 treated CSC that likely went undetected (figure 1A); nevertheless, this apparently small (but likely focal) decrease in endogenous FFA correlated with a significant reduction in the extent of fusion in the CSC fusion assay (Fig 1). As will be detailed below, this is why we work with isolated CV  -  it is difficult to draw effective molecular-functional correlations with CSC; rather, we simply use them as a first-pass analysis to look for effects on function, which is also why we used three different PLA2 inhibitors in initial experiments, so as not to miss potential correlations.

3.              In Figure 4: The authors are using human antibodies to identify sea urchin proteins. To univocally identify PLAin the 2-D gels, a mass spectrometric analysis of the regions where the western blot analysis indicated a PLAspot is desired. A mass-spectrometric detection, in this case, would make the conclusion of the paper significantly stronger. Can the authors comment on the feasibility of such an experiment?

Response: Yes, attempts were made to identify urchin PLA2 isoforms using in-gel trypsin digestion of the Coomassie stained 2D-gel spots (i.e. those that were immune positive on the corresponding western blot), followed by LC/MS/MS. The immune positive spot detected by the sPLA2 antibody was identified as a 15.33 kDa uncharacterized urchin protein (uniprot ID: W4Y7N6) with –log p value of 61.32 and 2 unique peptides out of 3 peptides in total. Similarly, the immune positive spots detected by the iPLA2 antibody were identified as a 61.49 kDa uncharacterized urchin protein (uniprot ID: W4ZKG2) with –log p value of 75.19 and 5 unique peptides out of 5 peptides in total. The search was done using PEAKS software on a database downloaded from Uniprot (also containing unreviewed entries). Note that no peptides corresponding to conserved regions of PLA2 were found, and thus that these peptides were likely from a more abundant protein in the spots of interest. In this regard, it should also be noted that the immunoblotting protocol used has an established sensitivity into the low attomole range with highly specific antibodies, and into the low femtomole range even with cross-species detection (Coorssen et al. Anal. Biochem. 2002). It is also thus equally likely that peptides from a more abundant protein in the spots of interest (i.e. as defined by Coomassie staining after matching to the western image and aligning of MW markers) simply overwhelmed the peptides of the urchin PLA2 species and that the identified peptides are thus wholly unrelated. Overall, the most direct answer is that while ‘feasible’ the obvious analysis did not yield effective information.

4.       Figure 5, legends – the comment regarding the removal of lanes in the chromatogram should refer to figure 5B, not 4B.

Response: Thank you, corrected.

5.                  It is highly unclear, how the experiment presented in figure 1A and Figure 2 are different than the one in Figure 8 and figure 7, as the authors themselves state that CV-PM fusion is indistinguishable than CV-CV fusion. In fact, CV-PM fusion may be considered by many as ‘more physiological’. How does the experiment in Figure 7+8 provide the reader with additional/complementary information to that of Figure 1A+2? Please elaborate and consider grouping the two together.

Response: As CV are fully primed and fusion-ready, whether endogenously docked on the PM (i.e. CSC; CV-PM fusion assays / exocytosis in vitro) or isolated to high purity and studied in CV-CV fusion assays, the main point to consider is that CV retain the minimal essential machinery for docking/priming, calcium sensing, and fusion. Hence why CV-CV fusion is indistinguishable from CV-PM fusion. However, in terms of trying to dissect underlying molecular mechanisms, despite CSC being a ‘more physiological’ preparation, the PM also constitutes a significant ‘background’ to any molecular analyses. That is, the PM (and other associated membranes; see above) also contains myriad proteins and lipids that have no (direct) role(s) in the late steps of regulated exocytosis. For years we have thus used this approach: establish whether a protein or lipid of interest can be linked to the late steps of exocytosis using CSC, then use the CV preparations and the CV-CV fusion assays to home-in on and do the ‘fine dissection’ of the probable roles of the molecules in the physiological fusion machine and/or the fundamental fusion mechanism (Churchward et al. Biochem. J. 2009). Furthermore, CSC fusion assays are somewhat limited, essentially enabling only a measure of select parameters (i.e. fusion extent and Ca2+ sensitivity) that occur downstream of docking and priming. The CV-CV standard fusion assay is comparable in that regard whereas, importantly, the CV-CV ‘settle’ fusion assay provides information concerning vesicle docking and/or priming. Therefore, the more detailed studies with isolated CV enable quantitative assessment of molecules on the membranes that are fusing (i.e. without the extensive molecular ‘background’ of the PM) and thus to tightly couple the molecular and functional analyses. We have been using this approach for >20 years now and it has always been respected as a critical approach to defining the roles of key molecular players. Thus, here, the use of CSC confirms a likely role of PLA2 in the late steps of exocytosis, while the CV assays enable quantitative assessments focusing specifically on those critical membranes / the organelle specialized for release. As in all our previous research, clearly this approach has helped in drawing a coherent and novel conclusion that would not have been the case if CSC alone were used. Respectfully, we have thus left the presentation as is considering that the logic of this approach has been upheld now in numerous publications over the years. 

6.                  Regarding Figure 6 and the effect of iPLA2– How do the authors settle the controversy between figure 6A (no effect) and 6C? How do the results in figure 6A can be settled with the large effects of BEL and FK on fusion (Figure 2, 7) and with the large effect on FFA production (Figure 1B, 8)? Why is PED6 suddenly being used as substrate? And how do vesicles with a trypsinized coat still maintain 50%-100% Ca2+-dependent fusion ability?

Response: Fig 6A is clearly noted to show iPLA2 activity on isolated CV membranes (i.e. suspended in BIM, as fully described in the Methods), whereas 6C shows activity on the surface of intact CV. The difference in the observed activities is most simply explained by the possibility that some iPLA2 were likely lost during the isolation of the CV membranes. After recognising this possibility, further experiments were designed to assess intact CV using the PED6 substrate, to fully enabled direct and selective measurement of PLA2 activity. Ablation of CV surface proteins to study function has proven quite informative in the past (Coorssen et al. J. Cell Sci. 2003; Coorssen et al. J. Biol. Chem., 2003); indeed, the senior author and collaborators designed and fully capitalized on those original studies. Why fusion ability is retained is still an open question, but the original results did rule out certain key proteins as ‘drivers’ of fusion and suggest linkage of others to calcium sensing. The data also emphasize the importance of non-proteinaceous components, and hence much of our work over the last several years on lipids and the enzymes locally/focally generating them. Nonetheless, more extensive trypsinization results in a complete loss of fusion (established ~30 years ago); that is, no nonspecific fusion (i.e. induced only by calcium interacting with lipids) remains either. Thus, such ablation of surface proteins serves as an important tool is establishing functional associations. However, returning to and focusing on the current results, the important message here is that treated CV still showed 100% fusion in the standard assay but aliquots of the same CV suspensions showed an ~50% decrease in docking and/or priming efficiency that correlated with the loss of iPLA2 (as assessed by western blotting and enzymatic activity).

7.                  Line 555 ‘.’ is missing after ‘(Fig. 3B)’.

Response: Line 556 ‘;’ is used after (Fig. 3B)’.

In summary, throughout the paper, the data collected as evidence for the sPLAexpression and effect are rather consistent and convincing – from the effect of LY on FFA production and fusion, to the strong signal at the 2-D gel, and to the results presented in figure 5, using a luminal CV fraction. In contrast, the results regarding the expression of iPLAare less convincing, and should be further clarified.

Response: The significant accumulation of TAG in the BEL treated CV and CSC (Supplementary Fig. 1) adds further support to the presence of catalytically active iPLA2, consistent with the identification by western blotting (Figure 4C and 6B) and the decreased PLA2 activity in trypsin treated CV (figure 6C). The proposed ‘housekeeping’ function of iPLA2 is to generate lyso-phospholipid acceptors, inhibition of the enzyme thus leads to increased FFA levels; under such conditions, the bulk of FFA is incorporated into TAG (Supplementary Figure. 1A, B).  Removing iPLA2 from the membrane or selectively inhibiting it with BEL or FK resulted in a loss of activity that correlated with a decline in docking/priming. We hope this is satisfactory to the reviewer.